# AI Influence: Mechanisms, Amplifiers, and Consequences

## Abstract

AI influence refers to AI's impact on the knowledge and values of individuals by acting as producers, mediators, and receivers of information. As a result, it impacts our collective processes of creating and spreading knowledge, forming beliefs, and reaching consensus. We argue that there are mechanisms of inconspicuous influence in AI development and deployment pipelines, which, when amplified by societal dynamics, could lead to dangerous outcomes that we may reverse by early interventions. We detail those mechanisms, amplifiers, and potential long-term consequences.

## 1 Introduction

### 1.1 Overview of AI Influence

**AI influence** refers to the impact of AI technologies on the knowledge and values of individuals, whether as a producer (e.g., LLM output), mediator (e.g., recommender system), or receiver (e.g., preference learning from human feedback) of information. As a result, it further impacts our collective processes of creating and spreading knowledge, forming beliefs, and reaching consensus. Different from "AI's impact in general", which concerns AI's broad societal impacts encompassing economic, legal, social, and environmental dimensions, "AI Influence" specifically zeroes in on the epistemic and axiological dimensions of this impact. It is concerned with how AI alters how humans know (epistemology) and what humans value (axiology), rather than just the direct outcomes of AI deployment.

**We propose "AI influence" to unify scattered research efforts.** Empirical research on AI influence is ongoing but scattered. Those efforts are either clustered around specific affected subjects — Wikipedia (Wagner & Jiang, 2025), Stack Exchange community (Burtch et al., 2024), open-source community (Yeverechyahu et al., 2024), scientific publication and peer review (Liang et al., 2024a;b), political campaigns and elections (Hackenburg & Margetts, 2024a; Potter et al., 2024) — or carved up along discipline boundaries like machine learning, cognitive science, education, human-AI interaction, and epistemology, with little cross-disciplinary discourse taking place.

**AI influence is not necessarily a harm.** Despite that AI influence on human epistemology poses serious concerns, it is too early to conclude that AI influence is, on net, a bad thing. Humans are bound by cognitive limitations, and it's likely that AI may expand our cognitive capacity and improve our collective deliberation.

### 1.2 Our Contributions

**Proposing AI influence as a distinctive research field** AI influence concerns the pervasive, subtle, and long-term ways AI reconfigures human internal states — in particular, the cognitive processes of acquiring and using knowledge, forming beliefs, and making judgments. This is distinct from AI safety or AI ethics research, where researchers address direct harms imposed on marginalized groups or engineering failures of AI systems (Gross, 2023; Hendrycks & Mazeika, 2022).

**Introducing a three-level framework for AI influence** We decompose the landscape of AI influence into three dimensions: *mechanisms*, the basic channels through which AI influences human epistemology; *amplifiers*, external factors that significantly enlarge such influence; and *consequences*, societal hazards that

$$\theta_{t+1} \mid a_t, \theta_t \sim \mathcal{D}(a_t, \theta_t)$$

**Mechanisms**: Sources of immediate impact of **AI actions** on human beliefs/preferences

**Amplifiers**: Factors that escalate impact on the **previous human belief/preference** to the next time step

**Consequences**: **Long-term change** in human beliefs/preferences due to earlier AI actions.

Figure 1: Formalizing the distinction between mechanisms, amplifiers, and consequences.

the amplified influence have led to or may soon lead to. These are *orthogonal* dimensions often connected by fully interconnected relationships, which allows us to focus on each of them individually while also touching on their connections.

**Summarizing methodologies for studying AI influence** We summarize all the methods that are used so far to study AI influence in Table 2. We also reason from first principles what could be used to study the influence from technologies on human truth-seeking and morality development process. We illuminate the gap and discuss future research directions.

| | Language Models (Supervised) | Language Models (Reinforced) | Recommender Systems | Knowledge-Based Systems |
|---|---|---|---|---|
| **Context Space** $\mathcal{C}$ | Natural-language prompts | | User-item pairs | Assertions/attributes |
| **Action Space** $\mathcal{A}$ | Natural-language responses | | Scores/rankings | (Truth) Values |
| **Distribution** $\mathcal{D}_{\mathcal{C}}$ | Training-time distribution of contexts | | | Expert input coverage |
| **Policy Space** $\Pi$ | Parameterized deep neural networks | | | Deterministic assignments |
| **Loss Function** $\mathcal{L}_\theta$ | Cross-entropy loss against human response | Plackett-Luce disagreement with human preference | Disagreement with human scoring/ranking | Abidance with expert-sourced constraints (0/1) |
| **Regularizer** $\Omega$ | Weight decay | KL regularization | Social regularization | Logical consistency constraints (0/1) |

Table 1: How four example types of AI system training/development can be mapped onto the common formalism. Each column represents one simplified, canonical example of that type of system, with only distinctive features shown.

## 2 Absence of AI Influence in Existing Paradigms

A large class of intelligent systems are designed to interact with humans in ways that alter human cognition. Current examples of such systems include:

- **Language Models and Language Agents.** Language models undergo pretraining on massive amounts of human-generated data (Devlin et al., 2019), which equip them for interaction with human users on information-seeking tasks. More importantly, the alignment training on these models (Ji et al.) through reinforcement learning from human feedback (Bai et al., 2022), direct preference optimization (Rafailov et al., 2023), or personalization methods (Chen et al., 2024), makes them produce user-preferred behaviors in explicitly interactive setups.

- **Recommender and Ranking Systems.** These systems function as information gatekeepers that filter vast item spaces to present users with prioritized content. Architectures in this domain have evolved from matrix factorization techniques (Koren et al., 2009) to deep candidate generation and ranking models (Covington et al., 2016). While typically framed as predictors of static user preferences based on historical interaction data, modern iterations frequently employ reinforcement learning objectives to maximize long-term engagement rewards (Chen et al., 2019). This optimization process treats the user's internal state as a dynamic component of the environment. Consequently, the system actively shapes user preferences by determining the exposure distribution of information and altering the choice architecture available to the agent.

- **Knowledge-Based and Decision Support Systems.** Knowledge-based systems often utilize structured data representations, including knowledge graphs and ontologies (Wang et al., 2017), or hybrid neuro-symbolic architectures that integrate deep learning with rule-based constraints (Belle, 2020). Unlike open-ended generation, these tools operate within strictly defined action spaces to output risk scores, diagnostic suggestions, or factual retrievals. The epistemic influence here manifests through automation bias and anchoring (Bansal et al., 2021). By presenting calculated probabilities or retrieved facts with high system confidence, these tools establish authoritative baselines that shift the human decision boundary regardless of the underlying ground truth.

We focus on this class of AI systems in the rest of this paper and avoid distinctions between them, as their commonalities induce the mechanisms, amplifiers, and consequences that we will introduce. In this section, we first dissect the currently mainstream formalisms that guide the training of these systems, and then specify the categories of missing elements.

## 2.1 Common Formalism for Learning From Humans

One common pattern emerge upon examining the mainstream formalisms in the development and training of all major types of AI systems that we consider. This high-level pattern involves a space $\mathcal{C}$ of *contexts*, a distribution $\mathcal{D}_\mathcal{C}$ over contexts, a space $\mathcal{A}$ of *actions*, and a space $\Pi \subseteq \Delta[\mathcal{A}]^\mathcal{C}$ of *policies* mapping any context to a distribution over actions. The task is then to solve the optimization problem

$$\text{minimize } \mathrm{E}_{c \sim \mathcal{D}_\mathcal{C}} \left[ \mathcal{L}_\theta(c, \pi(c)) \right] + \Omega(\pi), \quad \text{s.t. } \pi \in \Pi,$$

where $\mathcal{L}_\theta(\cdot, \cdot)$ is a (parameterized) loss defined against human annotation, and $\Omega(\cdot)$ a regularization term.

Table 1 explains how the training or development of the aforementioned types of systems — language models (Devlin et al., 2019; Bai et al., 2022), recommender systems (Resnick & Varian, 1997), and knowledge-based systems (Akerkar & Sajja, 2009) — can be mapped to such a formalism. It also implies what the parameter $\theta$ stands for, i.e., a human *belief state*, describing the beliefs, preferences, etc. of the human in the training loop.

There is the further dimension of *time*. The common formalism above is typically used to accomodate time in one of two ways.

1. *Assuming Full Stationarity.* When the loss function and the context distribution is stationary and the policy is memory-less, the interaction at each time step is simply a replay of the previous time steps. As such, the temporal setup can be directly reduced to the one-off setup. This is the case in, for example, language model training (Bai et al., 2022; Hadfield-Menell et al., 2016).

2. *Assuming Stationarity of Loss Function.* In formalisms such as Markov decision processes, the context (state) depends on the interaction history, and the policy is memory-ful and maps a pair of context *and history* to an action distribution. However, they still assume a stationary loss (reward) function, i.e., $\theta_t = \theta$, for some fixed reward parameter $\theta$. This is the case in, for example, most recommender systems trained with reinforcement learning (Afsar et al., 2022).

In reality, however, neither approaches fully suffice, as the implicit assumption of a stationary loss function ($\theta_t = \theta$) fails. It fails due to the natural shift and development of human beliefs/preferences, and, importantly,

due to the presence of *AI influence.* In other words, the loss function (parameterized by $\theta_t$) at the $t$-th time step is often a function of the previous actions taken by the AI system.

Note that $\theta_t$ can either be the state of a single human source (e.g., a user convinced of conspiracy theories who now gives conspiracy theory-aligned preference feedback), or that of a human collective (e.g., a company's hiring distribution being biased by a discriminatory hiring decision support system, which feeds back into the system's training data).

In the following subsections, we examine this missing influence in the training paradigms, from three precisely defined angles (Figure 1).

### 2.2 Mechanisms: Sources of Immediate Influence

Figure 1 shows how the loss function $\mathcal{L}_{\theta_{t+1}}$ (decided by the preferences and beliefs of the humans that the system learns from) at time step $t + 1$ depends on both the previous time step's human preferences and beliefs ($\theta_t$) and the action taken by the AI system ($a_t$).

In Section 3, we will qualitatively explore the direct *mechanisms* of AI influence, i.e., the causal pathways through which the AI system's action $a_t$ directly and immediately impacts the human belief state $\theta_{t+1}$ at the next time step. Ranging from persuasion (Durmus et al., 2024) to reliance (Nirman et al., 2024), such impact is empirically well-established through human studies (Jakesch et al., 2023; Glickman & Sharot, 2024b) and has been theoretically studied in a reinforcement learning setup (Carroll et al., 2024).

### 2.3 Amplifiers: Temporal Factors that Escalate Impact

In contrast to mechanisms, we define *amplifiers* as the factors that lead to the escalation of AI impact on the human belief state from the previous time step ($\theta_t$) to the next ($\theta_{t+1}$), i.e., the causal dependence of $\theta_{t+1}$ on $\theta_t$ that results in compounding errors.

Such factors can be endogenous, such as human confirmation bias (Oeberst & Imhoff, 2023), or exogenous, such as echo chambers formed by human-AI interaction (Glickman & Sharot, 2024b; Sharma et al., 2024), or institutional factors that entrench biased consensus (Lawrence et al., 2001; Bisson et al., 2021). We will discuss these factors in more detail in Section 4.

### 2.4 Consequences: Impact on Belief States Over Time

When both mechanisms of influence and amplifiers of influence are present, they tend to imply that the impact of AI actions on the human belief state $\theta_t$ compounds over time, eventually leading to large and potentially irreversible changes. Again, such changes can either be at the individual levels (beliefs, preferences) or at the collective level (norms, culture, collective knowledge).

In Section 5, we qualitatively characterize some key potential consequences of amplified AI influence. Some of them have already seen strong empirical evidence, while others are currently speculative predictions about long-term outcomes.

## 3 Mechanisms

In this section, we cover specific mechanisms through which AI systems play a direct and immediate role in influencing human epistemics and morality, at an individual level and societal level. By *mechanisms*, we refer to either technical limitations of AI systems or new ways through which humans interact with AI that directly causes change in human epistemics.

Here, we emphasize that the AI systems change how information is originated, disseminated, propagated, and received by humans or AI systems. The scope extends beyond that of algorithmic biases.

Table 2: Related research classified by methodology and topic. Empty cells indicate the lack of known works. Due to the relative scarcity of qualitative studies, we include them as a single category while using a fine-grained partition for quantitative studies.

| | | Qualitative Research | Formal Models | Simulations | Descriptive Analysis | Observational Causal Inference | RCTs |
|---|---|---|---|---|---|---|---|
| **Mechanisms** | Digital Reliance | Gerlich (2025); Hirvonen et al. (2024); Glickman & Sharot (2024a); Brandtzaeg et al. (2024); Kulveit et al. (2025); Kirk et al. (2025) | | | Gerlich (2025); Burtch et al. (2024); Nirman et al. (2024); Thompson et al. (2024); Wagner & Jiang (2025); Dillion et al. (2025) | Burtch et al. (2024) | Kruegel et al. (2025) |
| | Distinct AI Biases | Köbis et al. (2021); Brandtzaeg et al. (2024) | Taori & Hashimoto (2022) | Haroon et al. (2023) | Adilazuarda et al. (2024); Barman et al. (2024); Lamparth et al. (2024); Ryan et al. (2024); Agarwal et al. (2025); Weng et al. (2025); Wang et al. (2025) | Brown et al. (2022); Haroon et al. (2023); Yakura et al. (2024) | Glickman & Sharot (2024b); Fisher et al. (2024); Danry et al. (2024); Costello et al. (2024a); Kidd & Birhane (2023b); Kruegel et al. (2025); Leib et al. (2021); Piccardi et al. (2024); Potter et al. (2024) |
| | Direct Persuasion | Brandtzaeg et al. (2025); Hirvonen et al. (2024) | Collins et al. (2025); Dean et al. (2024) | Ferraro et al. (2024) | Fisher et al. (2024) | Hosseinmardi et al. (2024) | Argyle et al. (2025); Hackenburg & Margetts (2024a); White et al. (2024); Goel et al. (2025); Costello et al. (2024a); Durmus et al. (2024); Matz et al. (2024); Danry et al. (2025) |
| | Attention Reallocation | Schuster & Lazar (2025) | | | | Su et al. (2016) | Mendler-Dünner et al. (2024b); Haupt et al. (2023); Hosseinmardi et al. (2024) |
| **Amplifiers** | Human-AI Dual Influence | Brady et al. (2023); Brady & Crockett (2024); Collins et al. (2024); Lazar et al. (2024); Li & Yin (2024); Pedreschi et al. (2025) | Lin et al. (2024); Ferbach et al. (2024); Collins et al. (2025); Qiu et al. (2025); Krueger et al. (2020) | Wang et al. (2024); Brinkmann et al. (2022); Ferraro et al. (2024); Collins et al. (2025); Qiu et al. (2025); Mansoury et al. (2020); Perra & Rocha (2019); Dean et al. (2024) | Li et al. (2023); Liang et al. (2024a) | Qiu et al. (2025) | Glickman & Sharot (2024b); Brinkmann et al. (2022); Chan et al. (2024); Pataranutaporn et al. (2023); Haupt et al. (2023); Hosseinmardi et al. (2024); Lu et al. (2024); Pappalardo et al. (2024); Sharma et al. (2024) |
| | Trust | | | | Dillion et al. (2025); Araujo et al. (2020); Helberger et al. (2020) | | Narayanan et al. (2023); Pataranutaporn et al. (2023); Reis et al. (2024); Osborne & Bailey (2025) |
| | Institutional Path Dependence | Kulveit et al. (2025); Simon & Isaza-Ibarra (2023); Aoki (2024); Gruetzemacher et al. (2024); Lazar & Manuali (2024); Matz et al. (2024); Ovadya et al. (2024); Leibo et al. (2025) | Jarrett et al. (2025) | Zhang et al. (2025); Jarrett et al. (2025) | | | Potter et al. (2024) |
| | Socio-Economic Matthew Effect | Capraro et al. (2024) | | | Wang et al. (2025) | Su et al. (2016) | |
| **Consequences** | Lock-in of Human Errors | | Collins et al. (2025); Qiu et al. (2025); Lin et al. (2024) | Qiu et al. (2025); Collins et al. (2025); Wang et al. (2024); Mansoury et al. (2020); Perra & Rocha (2019) | | Qiu et al. (2025) | Chan et al. (2024); Costello et al. (2024a); Haupt et al. (2023); Kubin & Sikorski (2021) |
| | Lock-in of AI Biases | Brandtzaeg et al. (2024); Kulveit et al. (2025); Köbis et al. (2021) | Taori & Hashimoto (2022) | Piao et al. (2025) | Adilazuarda et al. (2024); Barman et al. (2024); Lamparth et al. (2024); Ryan et al. (2024) | | Glickman & Sharot (2024b); Fisher et al. (2024); Danry et al. (2024); Costello et al. (2024a); Kidd & Birhane (2023b); Kruegel et al. (2025); Leib et al. (2021); Piccardi et al. (2024); Potter et al. (2024) |
| | Value Capture | Nguyen (2024b) | | | | | |
| | Knowledge Collapse | Brandtzaeg et al. (2024); Glickman & Sharot (2024a); Koskinen (2024); Wihbey (2024) | De Marzo et al. (2024) | Peterson (2024); Bossens et al. (2024) | Burtch et al. (2024); Dohmatob et al. (2024); Thompson et al. (2024); Li et al. (2023); Liang et al. (2024a); Si et al. (2024); Wagner & Jiang (2025); Wu et al. (2024) | Anderson et al. (2024) | Doshi & Hauser (2023); Padmakumar & He (2023); Sharma et al. (2024) |
| | Epistemic Stratification | Kay et al. (2024) | | | Agarwal et al. (2025); Wang et al. (2025) | | |

### 3.1 AI Introduces Distinct Biases into Collective Knowledge

Although AI systems are trained on data generated by humans, they do acquire distinctive biases from humans (Glickman & Sharot, 2024b; Kahneman et al., 2021). Specifically, there are the following reasons that introduce distinctive AI biases:

- *Learning systems like LLMs are struggling with long-tail knowledge.* As a primary example of learning-based intelligent systems, the question-answering accuracy of LLMs correlate strongly with how many times questions and answers co-occur in the training dataset (Kandpal et al., 2023; Das et al., 2024).

- *Architectures create unique AI biases.* Architectural biases often stem from technical limitations, as opposed to biases in datasets that can be more readily resolved by more training or more data. One notable example is the bias of Convolutional Neural Networks (CNNs) towards texture (Geirhos et al., 2018). Tokenization, the strategy LLM employs to split words into subwords, introduces biases unique to AI, such as downgrading arithmetic performances (Singh & Strouse, 2024), mishandling grammatical structures, and biases in handling rare words (Phan et al., 2024).

Through training and deploying AI systems that acquire distinct biases, we risk introducing new biases into the collective knowledge-making process, such as publication, journalism, scientific research, etc. Such AI biases might be persistent or even amplified because of digital reliance or feedback loops, as we will discuss in the following two subsections.

### 3.2 Cognitive Offloading, Cognitive Enhancements, and Digital Reliance

AI can enhance human cognitive performance, which can take place either directly by providing advice and implementable solutions (Senior et al., 2020; Fawzi et al., 2022) or indirectly by revealing novel cognitive strategies and problem-solving approaches (Shin et al., 2023). Cognitive offloading is the term commonly used to describe such activities, namely, physical actions (such as preparing a grocery list) to reduce cognitive demands required (Risko & Gilbert, 2016). Research shows that humans are willing to offload attention-demanding tasks to AI systems (Wahn et al., 2023). AI systems are also used to improve human cognitive performance. For example, a study that examines the performance of Go players (Shin et al., 2023) reveals that the performance of Go players improved after being exposed to AlphaGo moves, possibly as a result of learning novel non-human strategies from AlphaGo. Consistent results come from a study examining human problem-solving in a navigation task (Brinkmann et al., 2022). In this study, participants navigated through complex networks. Each path was associated with rewards (earning points) or penalties (losing points). Before performing the task, participants were exposed to solutions generated by the AI or by humans. The results demonstrated enhanced performance (accumulation of higher rewards) among players learning from AI, mainly due to the exposure to counterintuitive but optimal strategies generated by the AI. For example, the AI better identified than humans paths that initially appeared suboptimal but ultimately yielded better outcomes.

On the other hand, those cognitive offloading and enhancement activities enabled by AI may lead to digital reliance. Research demonstrates that reliance on digital tools, and in particular AI, alters different cognitive processes such as memory, critical thinking, and problem-solving. For example, Sparrow et al. (2011) showed that when information is accessible through search engines, individuals prioritize remembering where to find this information rather than retaining it. This pattern extends to modern AI systems as well. Gerlich (2025) found that cognitive offloading to AI tools correlates with reduced critical thinking engagement, particularly among younger users who exhibit higher dependency. Consistent with these empirical findings, Zhai et al. (2024) conducted a systematic review revealing that over-reliance on AI dialogue systems impairs critical thinking and decision-making by fostering cognitive shortcuts. Together, these studies suggest that in some cases, delegating cognitive tasks to AI systems may deteriorate fundamental cognitive and thinking capabilities.

In the context of this paper, digital reliance makes space for bias amplification, as we will discuss in the following subsections.

### 3.3 AI Persuasion Directly Reshapes Human Beliefs

As AI systems become integral to how humans access and evaluate information, they exert increasing influence over the processes of belief and opinion formation. Recent studies demonstrate that conversational AI can measurably shape political attitudes (Hackenburg & Margetts, 2024b; Fisher et al., 2025) and alter support for electoral candidates (Argyle et al., 2025). While persuasive capacity generally scales with model size, post-training methods and prompting strategies can yield even larger effects (Durmus et al., 2023; Hackenburg & Margetts, 2024b). Consequently, smaller models with targeted fine-tuning can achieve persuasive capabilities comparable to frontier systems, rendering influence tools broadly accessible. Techniques that maximize persuasive effectiveness, such as information-dense rhetoric, are associated with systematic reductions in factual accuracy, indicating a potential trade-off between persuasive power and epistemic reliability (Hackenburg & Margetts, 2024b).

Although such persuasive capabilities raise concerns about manipulation for financial or political gain, they may also be directed toward prosocial ends. Targeted human–AI dialogues have been shown to increase effective charitable giving beyond either static AI messages or human appeals (White et al., 2024). Similarly, conversational interventions with AI can durably reduce conspiratorial thinking, with effects persisting for up to two months (Costello et al., 2024b), and can decrease confidence in false beliefs (Goel et al., 2025). At the individual level, the epistemic influence of AI can be seen as dual-use: it can amplify both epistemic risk and epistemic improvement, depending on the underlying objectives.

### 3.4 AI Reallocates Human Attention

One of the major functionalities of AI systems is that they reorganize and redistribute information available to us, as search engines (including LLM-based ones) and RecSys-based social media do. In the previous subsection, we cover new mechanisms through which AI biases affect human judgements, while in this case, AI influences what we see and think by selecting what information gets presented to us and receives our attention. This may have a strong agenda-setting effect on our thinking (Mendler-Dünner et al., 2024a).

We elaborate on the problem of attention allocation and the resulting segmentation of users. For sophisticated users of AI technologies, it is possible for generative models to be hugely creative, adding to intellectual diversity (Meincke et al., 2024). But such possibilities require careful technique and strategy, from few-shot prompting to chain-of-thought reasoning to iterative strategies in general. For the vast majority of the model-using public, who may not understand what the models are and do, and have little ability to execute prompt engineering strategies, usage may be largely passive and simplistic. Models will therefore tend to provide answers and content to the majority of users that conform to mainstream, modal patterns — the most likely next token, the probabilistic best answer or idea. This, in fact, is their central tendency and what they are designed to do. Using the models in a simplistic auto-complete or recommendation engine-style is likely to direct human attention to mainstream ideas and trends that are featured prominently on the open web (where the model pre-training has taken place), and not necessarily to more diverse, challenging, obscure, or marginal ideas or viewpoints.

## 4 Amplifiers

Mechanisms enumerated in Section 3 explain the forces that AI systems exert on human cognition and epistemology. Those forces tend to be subtle and may not pose extreme risks on their own.

Meanwhile, in this section, we introduce a range of *amplifiers* that are *external* to AI systems and may significantly increase AI influence (usually social factors), to the degree of posing systemic risks described in Section 5.

### 4.1 Human-AI Dual Influence Creates Feedback Loops

The influence between AI and humans is not one-directional. Humans' preferences can be influenced by the content generated by AI systems, while AI systems are trained to align with human preferences as well (e.g., Reinforcement Learning with Human Feedback (Ziegler et al., 2019)). Such a feedback loop between humans

and AI is similar to the feedback loop between content users and content creators in recommender systems, where users' tastes are shaped by the content they consume and creators produce content to fit users' tastes (Jiang et al., 2019; Lin et al., 2024).

Although human-AI dual influence might help to improve the alignment between humans and AI, it could also bring potential harm. For example, when humans or AI have initial biases or errors regarding a certain topic, such biases and errors can be circulated and amplified in human-AI interactions. There has been extensive research on human-to-AI and AI-to-human influence, but it was not until very recently that research showed human-AI interaction may further exacerbate this influence mechanism: biased AI systems can affect human beliefs, rendering humans more biased compared to the initial state, due to the amplification of bias by AI systems and assigned trust by humans in AI judgments (Glickman & Sharot, 2024b;a).

AI bias is an established research field (Mayson, 2018). In this paper, however, we argue that digital reliance on AI and feedback loops established in human-AI interactions legitimize larger concerns over this topic. Not only because bias affects the accuracy of medical decisions (Challen et al., 2019) or racial fairness (Salinas et al., 2023), which are by themselves important problems, but also because those biases are permanently introduced into epistemic processes and alter our worldviews (Vicente & Matute, 2023).

## 4.2 Trust Amplifies AI Influence

Do higher levels of trust in AI correlate with increased AI influence? Recent research provides evidence supporting this claim. For example, Vicente & Matute (2023) demonstrated that higher trust in AI systems in medical diagnostic tasks led participants to adopt more of AI's biased recommendations, and even carry these into subsequent tasks. Similarly, it was found that self-reported trust in AI systems was associated with the persuasiveness of deceptive AI classifications; interestingly, trust was not associated with the effect of improved AI-generated truthful explanations in the case of news headlines (Danry et al., 2024), although results to the contrary were found in a credit loan decision-making setup (Sunny, 2025).

Current evidence suggests that human trust in AI is highly sensitive to context and culture. While in many contexts, people prefer AI advice over humans' (Araujo et al., 2020; Logg et al., 2019), in high-stakes contexts (such as medicine or other life-threatening cases), people assign trust to humans more than AI systems (Reis et al., 2024). Additionally, Globig et al. (2024) found that trust in AI varies significantly across cultures (Globig et al., 2024). Individuals in Eastern countries (e.g., India, Indonesia) exhibit greater trust and optimism towards AI compared to their Western counterparts (e.g., U.S., Germany), who tend to be more skeptical and cautious (Globig et al., 2024).

## 4.3 Institutional Path Dependence

Institutional path dependence refers to the tendency of organizations and systems to make decisions and adopt practices based on past trajectories, often locking in early patterns of behavior (Page et al., 2006). Epistemic frameworks through which institutions understand and address issues can be influenced by AI, an influence that can be hard to remove given the self-reinforcing nature of institutions (Arthur, 2018).

For instance, widespread AI application in the education sector may plant deep-rooted AI influence in children (Xu & Ouyang, 2022), AI advisors and analytics may bias governmental decision-making processes toward specific data-driven perspectives (Castelnovo & Sorrentino, 2021), AI-influenced public opinion can reinforce or challenge institutional norms (Panait & Ashraf, 2021), and early critical attitudes toward AI-generated art and writing have led to the enactment of institutional policies against the use of language models (Takagi, 2023; Kreitmeir & Raschky, 2023). Once these AI-mediated epistemic influences take root, their self-reinforcing nature may make it difficult to shift away from initial decisions, even in light of new evidence or changing contexts.

The self-reinforcing nature of the institutional path dependence problem will be particularly difficult to mitigate, given recursion (Peterson, 2024). Once embedded narratives take hold and the climate of human opinion gets expressed at scale on social media and the web, AI models themselves will subsequently be trained on this new data containing AI influence. This "data coil" means path dependence becomes difficult to resist or reverse (Beer, 2022).

### 4.4 Socio-Economic Matthew Effect

Advanced AI systems threaten to dramatically amplify existing socio-economic inequalities through what we term the "AI Matthew Effect", whereby initial advantages in AI access and capability compound exponentially over time.

Specifically, AI Matthew Effect occurs when groups initially receiving more benefit from AI (*e.g.*, the wealthy, speakers of majority languages, those living in developed nations, those with access to GPUs, those working in fields where training data is more abundant) receive cascading benefits, and vice versa. An example is when biases against minority languages in LLMs shrink their user base who speak minority languages, which could further reinforce biases against minority languages due to under-representation.

This dynamic could manifest through several interconnected mechanisms:

**Productivity amplification:** AI systems act as force multipliers for human productivity, with their effectiveness scaling in proportion to the user's existing capabilities and resources. High-skilled knowledge workers with access to state-of-the-art AI tools can leverage them to augment their expertise, potentially increasing their productivity by orders of magnitude. Meanwhile, workers in lower-skilled positions may find their jobs automated or devalued, creating a widening productivity gap.

**Capital concentration:** Organizations with early access to powerful AI systems can optimize operations, reduce costs, and capture market share more effectively than competitors. This advantage creates a self-reinforcing cycle where increased profits enable further AI investment and development, leading to market concentration.

## 5 Consequences

Influence mechanisms (Section 3), whose effects are magnified by amplifiers (Section 4), may lead to long-term consequences that are associated with large-scale hazards.

Long-term consequences are hard to clearly demonstrate in advance, but some have nonetheless manifested in empirical studies. Here we make a non-exhaustive list of these potential consequences.

AI systems that are trained on human data contains human errors and biases (Mayson, 2018; Binz & Schulz, 2023; Yax et al., 2024). Direct and indirect interactions with those models can circulate those biases back to humans (Morewedge et al., 2023; Valyaeva, 2024). Furthermore, those human errors and biases can be amplified via human-AI interactions because humans may assign more trust in AI output than average humans (Logg et al., 2019). These psychological traits of humans and the training methods of learning systems (e.g. LLMs) raise concerns that those human errors and biases might be permanently preserved, amplified, and even locked into human society over the long run. The term "lock-in" refers to cases where values, beliefs, knowledge, and practices are introduced into human society, last for a long time, spread widely, assume a dominant memetic position in a population, are institutionalized (therefore hard to remove), and cause damage (Hendrycks & Mazeika, 2022).

### 5.1 Lock-in of AI Biases

AI bias has been well documented and studied — not only in the realms of fairness and equality (Bolukbasi et al., 2016; Caliskan et al., 2017), but also on broadly construed biases in cultural and factual domains (Santurkar et al., 2023).

However, a consequential effect has been largely overlooked: when humans interact with these biased systems, they internalize the systems' amplified bias and become more biased than they initially were (Glickman & Sharot, 2024b; Vicente & Matute, 2023). This bias amplification feedback loop relies on two key characteristics of AI systems: First, AI systems provide a higher signal-to-noise ratio compared to humans, consistently producing less variable outputs than human judgments (Kahneman et al., 2021). Second, in many domains, humans perceive AI systems as more capable and accurate than other humans (Logg et al., 2019), making them more receptive to AI influence or uncritically adopting AI biases. For instance, clinicians inherit

AI biases even after AI systems are removed (Vicente & Matute, 2023). These characteristics create a dynamic where even small initial biases can be rapidly adopted and magnified through human-AI interactions. Furthermore, this effect raises particular concerns for children, who have more malleable knowledge structures and may be more susceptible to AI's influence than adults (Kidd & Birhane, 2023a), raising the concern that such AI biases would be locked-in over generations.

## 5.2 Goodhart's Law and Value Capture

Human objectives are often operationalized into quantifiable metrics — for instance, research quality being quantified as citation counts, and idea quality being quantified as the number of retweets. In economics, Goodhart's law, *"when a measure becomes a target, it ceases to be a good measure,"* states that optimizing for a quantifiable proxy initially leads to improvement in the true objective, but beyond a certain point, such optimization often leads to (potentially catastrophic) degradation in the true objective (Goodhart, 1984).

An instance of Goodhart's law in the case of human values, *value capture*, happens when one mistakes quantified proxies for their much richer terminal values, and exclusively optimizes for the former instead, thereby losing the ability of personal deliberation on their values (Nguyen, 2024a).

AI has already been used in such quantification of objectives, for example, in social media (Anandhan et al., 2018); other similar uses of AI has also been proposed, including as arbiters for resolving human disagreement (Tessler et al., 2024) and human representatives for collective decision-making (Zhang et al., 2024). In all such cases, human actors may be incentivised, or are already incentivised (Lüders et al., 2022; Wolf et al., 2017), to optimize for the AI-defined objectives. If such optimization becomes the dominant concern of human participants — which is plausible given that AI products are often designed to be game-like and addictive (De et al., 2025) — value capture may steer people's values and objectives away from an ideal deliberative choice.

## 5.3 Knowledge Collapse

Knowledge collapse (Peterson, 2024) is defined as *the progressive narrowing over time of the set of information available to humans, along with a concomitant narrowing in the perceived availability and utility of different sets of information.* It is hypothesized to manifest as a "mode collapse" of collective knowledge in the human community, where long-tail information is lost while mainstream information is strengthened.

Peterson (2024) mainly focuses on unrepresentative data, lack of in-depth exploration during LLM inference, and algorithmic limitations of next-token prediction as the potential causes of knowledge collapse. Peterson (2024) argues that by making mainstream information more readily available, learning systems like LLMs shift attention away from long-tail information.

In addition to these concerns, we note that other mechanisms outlined in this paper, including, for example, dual influence (Lin et al., 2024), can similarly contribute to knowledge collapse. From a mechanistic angle, knowledge collapse and lock-in share many commonalities, most especially the reinforcement of existing popular ideas and the suppression of marginal ones.

## 5.4 Epistemic Stratification

Epistemic stratification is the unequal distribution of access to knowledge, resources, and cognitive tools across individuals or groups, leading to disparities in their ability to acquire, evaluate, and generate knowledge (Silva Filho et al., 2023).

AI may contribute to epistemic stratification by amplifying existing disparities, such as through unequal access to advanced AI tools, biased algorithmic recommendations that reinforce echo chambers, the prioritization of information access for privileged demographics (Kay et al., 2024), or the increasingly centralized control over AI development (Brynjolfsson & Ng, 2023).

# 6 Caveats and Counterarguments

## 6.1 AI Systems Have Negligible Influence on Human Cognition

Empirical evidence does not provide a holistic picture of AI's impact on human cognition. It is true that humans are becoming more reliant on AI systems for their tasks, but it is unclear whether having AI systems to process those tasks for humans would necessarily degenerate or enhance those cognitive capabilities of humans. At least, it is still unclear whether humans' navigation skills are compromised because of using GPS tools (Fricker, 2021; Jadallah et al., 2017).

Two questions are instrumental to understanding AI systems' impact on human cognition. For one, does digital reliance influence human cognitive skills that are directly replaced by corresponding AI capabilities (Teschke et al., 2013)? For instance, does the use of GPS hurt human navigation skills (Fricker, 2021)? The same question could be asked about other digital tools and human skills, such as calculators and arithmetic skills, machine translation tools and second-language acquisition. For the other, do the replaced domain-specific human skills undermine more general human cognitive capabilities? For example, do the undermined arithmetic skills hurt human general mathematical reasoning and problem-solving abilities (Geary et al., 2015; Hurst & Cordes, 2018)?

Without sufficient empirical evidence on how human cognition might be altered in the presence of new tools, especially AI systems, it is hard to firmly hold our position. Hence, an alternative view is, AI systems may have a negligible impact on human cognitive capabilities over the long term. One reason is that we do not understand the relationship between low-level domain-specific skills and high-level general capabilities. Replacing the former by AI may have little negative impact on the latter, in which case adequate tool use may actually enhance cognitive capabilities (Teschke et al., 2013).

## 6.2 Highly Parameterized AI Systems Are Less Biased and Error-Prone Than Humans Are

AI systems are biased (Jadallah et al., 2017) and error-prone (Zhou et al., 2024), as research has revealed, but so are humans. Besides those inductive biases that are introduced by specific architectures and training methods (Geirhos et al., 2018; Singh & Strouse, 2024), AI systems acquire their biases from training datasets and, by extension, from humans. Highly parameterized AI systems such as LLMs are less biased and error-prone than conventional machine learning models as they are more expressive, and techniques such as RAG help to consult external sources for truth validation (Gao et al., 2023). Meanwhile, it is also likely that state-of-arts AI systems may become even less biased and error-prone than average humans are. From the point of view of collective truth-seeking (such as conducting scientific research and collective deliberation), AI systems functioning as "shadow authors" to individual humans can be positive.

That being said, err should be on the side of being cautious. It is likely that AI biases, errors, and hallucinations become more elusive before they are removed (Zhou et al., 2024). Once they are hard to find for average users, commercial developers are much motivated to address those problems, creating persistent and even amplified biases and errors (Ren et al., 2024), which are precisely what we warn in this paper.

## 6.3 AI's Epistemic Influence Can Be Positive

In Section 5, we have detailed AI's long-term impact on human knowledge and values. Notably, they seem overwhelmingly negative. It is not our intention to present negative views only, but we are likely biased and limited in our perspectives. We want to raise attention on AI's epistemic influence and avoid the cascading effects over the long term, but we also want to acknowledge that we are far from having a holistic picture.

It is entirely likely the issues we have raised here can be addressed over time and people can become wise in using those tools. For instance, users, especially students and researchers, may acquire a critical lens of AI-generated content. Under the name of "AI literacy", students are taught to use, understand, and evaluate AI systems critically (Casal-Otero et al., 2023). Sufficient critical thinking skills, paired with AI systems' increasing reach and capabilities, may cultivate a generation of more informed learners and citizens who are more capable of participating in collective truth-seeking and deliberative processes.

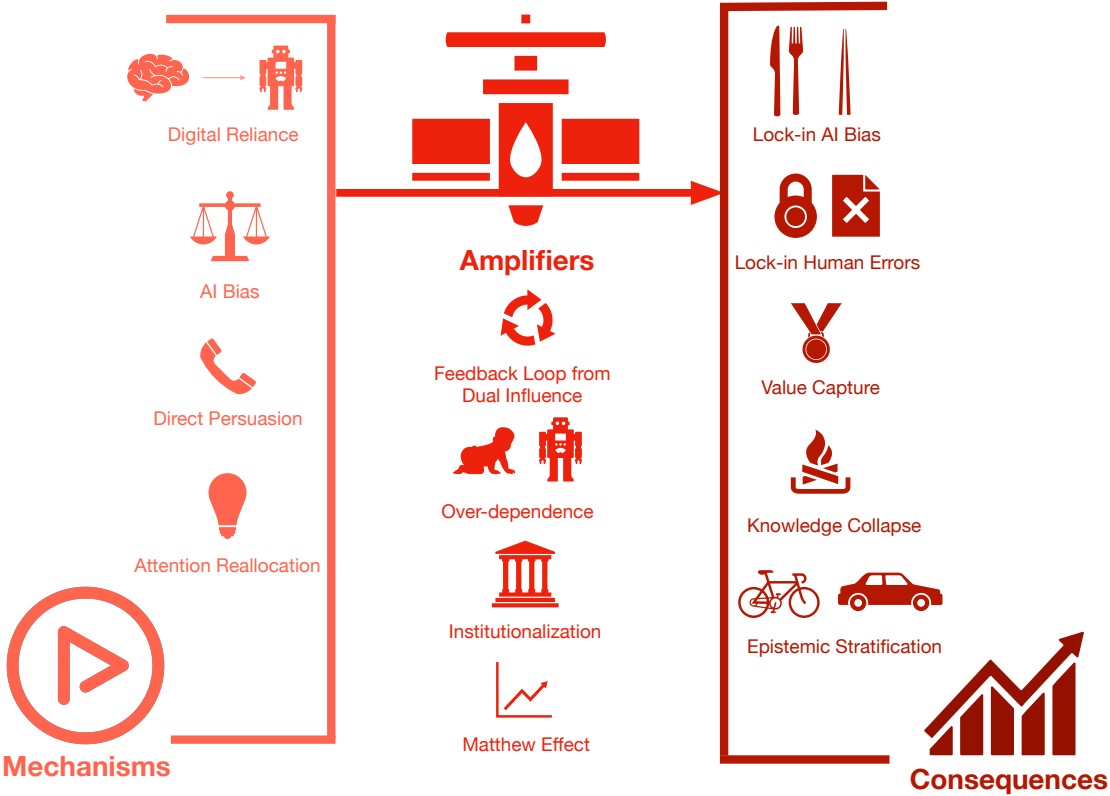

Figure 2: Mechanisms, amplifiers, and consequences of AI influence.

## 7 Conclusion

AI exerts systematic influence over the beliefs and values in individuals and society. We have outlined the mechanisms that enable such influence, the amplifiers that magnify the influence, and the potential consequences it may entail.

The eventual aim of AI influence research is to enable the responsible management of AI influence over human cognition, knowledge, and values, reaping its benefits while avoiding the harms. Accomplishing such an aim requires coordination between communities of interdisciplinary methodologies and perspectives, including AI safety and AI ethics communities, machine learning and human-computer interaction communities, social science communities, and, importantly, industry actors.

### Broader Impact Statement

Recognizing AI influence is a necessary precondition for managing it, and in this respect, we aim to promote societal interest by raising awareness on the issue. Since mid 2025, reports of epistemic and psychological harms from deployed AI systems — extreme examples of which include AI-driven psychosis (TREYGER et al., 2025), with milder examples of influence and reliance being exponentially more prevalent (Phang et al., 2025) — have become increasingly prevalent. Given the trend of increasingly wide and immersive deployment of AI systems, it is likely that such epistemic and psychological impact will expand by orders of magnitude in the near future. We hope that increased awareness on this class of problems can foster the development of technical and governance solutions for the management of AI influence.

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
