# OpenReview forum: "AI Influence: Mechanisms, Amplifiers, and Consequences"
_TMLR — Rejected by TMLR_

### Review · Reviewer_EsgC · 2025-11-21

**Summary Of Contributions:**

This paper fits the call of TMLR under the category "papers that contain surveys that draw new connections, highlight trends, and suggest new problems in an area." It is a position/survey paper that addresses three stages of how "AI" may influence people and societies, in both good and bad ways. The authors classify and cluster several existing phenomena, such as biases, reliance on tools, etc., into mechanisms, amplification, and consequences. For each phenomenon they bring arguments from a non-exhaustive list of related papers.

I see its main contribution by going one level higher above existing fields of fairness or bias in AI, perception of AI and human-AI collaboration, and possibly some others. It seems to me that the main objective is to bring several existing research directions under one umberalla and coin a new term "AI influence".

From the survey perspective, the paper is a great starting point to look into interdisciplinary research just beyond core ML. Table 1 provides a fine-grained typology - unfortunately the various methodologies (columns) are not mentioned and worked out in the text at all.

However, I'm having a hard time to get the main message of this paper from a research perspective. For one, it seems that it tries to bring several existing research directions under one umberalla and just introduce a new term "AI influence". But the question I'm asking is - why is this necessary in the first place? If "AI influence" is a superset of existing research on the surveyed phenomena (bias, and others), what new perspectives does the term bring on these problems?

The authors propose "AI influence" as a unified cross-disciplinary research direction. It's great to unite things and cross boundaries, but at the same time it seems rather obvious - if one's research is limited to only technical part (e.g., debiasing LLMs), the natural extension is to look beyond own field and reach to societal aspects or other disciplines.

I'm also missing any outlook what new problems can "AI influence" tackle, what solutions to the mentioned problems (the three stages) it can offer, or what are its main features worth adapting in a larger research community. Overall it reads like a position paper, but at the same time it does not really take any strong position worth arguing for. The authors promise "future research directions" but I do not find many new research avenues in the paper that spark a new research idea, direction, or open question (with small exception in section 5.1); the paper rather provides a static snapshot of the current state.

Overall, I cannot really pinpoint something that is objectively and conceptually wrong about this paper - all phenomena are well explained, with bringing up some relevant research. Yet it is missing in my view the major aspect of a (position) paper - why should we care? (I mean why should we care about coining new term and writing a paper about it; I absolutely agree that we should care about AI and its potential influence on society). There is no research question being asked and addressed, there are no arguments saying that the "AI influence" perspective is substantially distinct from other established research directions.

**Additional Comments:**

Suggestions:

Figure 1. If this summarizes the entire paper, could you maybe quickly describe these terms and remove these catchy but ambiguous icons instead? Because without knowing their exact meaning (which will comes in the text later, maybe), this graphic looks useless. It might make sense at the conclusion instead?

Typos:

"dataset. (Kandapal et" - extra full stop

"For example, A study" - capital A

"(Brinkmann et al. 2022) . In this study" - extra space

mixing dash, ndash, and mdash (-, --, ---)

**Audience:**

Yes

**Audience Explanation:**

I think the audience of TMLR is broad enough and might look for interdiscpilinary perspectives. Whether or not the TMLR audience will find the position/survey intriguing is a different question.

**Broader Impact Concerns:**

Does not apply, as this paper has an impact of AI as its subject.

**Claims And Evidence:**

No

**Claims Explanation:**

First, the paper does not specify the object of study, namely what it means by "AI". I'm fine with calling everything in machine learning just AI, but the problem is that the authors seem not to differentiate at all. Some references refer to specialized ML systems (e.g., AlphaGo), some are search engines (does it still qualify as AI?), somewhere LLM is used as a replacement for AI (2.1, first bullet point), and some are plainly wrong - for example the cited paper by Wahn et al. (2023) does not do anything even close to AI! It's just some "algorithm" to help counting moving objects in the simulated task demanding attention.

Second, there are claims without any empirical support, for example "Productivity amplification" and "Capital concentration" which seem to repeat some common wisdom that AI can boost productivity of experts. I'd expect critical evaluation of such claims, with citing relevant studies.

**Requested Changes:**

I think Table 1 is a really good starting point for a survey (synthesis and analysis) or a position paper (taking a stance and promoting an idea), and I believe an analysis along these two axes could actually bring value. It would be great to analyze the "AI influence" framework for each section, by each column, and to provide some synthesis why this pespective is helpful and what new research questions it can drive. I'm aware that what I'm asking for is to drastically rewrite the paper, though.

---

> ### Author Response · Authors · 2025-12-12
> **Summary of  Changes**
>
> Thank you so much for your feedback! In our newly revised manuscript, in addition to targeted improvements, we have made the following large structural changes.
> - Newly created **Section 2** that specifies the scope of systems we consider, formally explains the absence of AI Influence in the formalisms of these systems, and decomposes the impact of such missing influence into three components, each corresponding to mechanisms, amplifiers, or consequences. **In this new section, we have also aimed to unambiguously lay out what our "position" is.**
> - Newly created **Figure 1** that illustrates the precise and formal distinction between mechanisms, amplifiers, and consequences, and the reason why they form a complete picture.
> - Newly created **Table 1** that compares the formalism of language models training (supervised/reinforced), RecSys training, and the construction of knowledge-based systems. They serve as canonical examples in the scope of systems that we consider.
> - Moved "human-AI feedback loop" from mechanisms to amplifiers. The new formal distinction between the three components resolved the ambiguities and made it clear where it belongs.
>
> We have also made targeted improvements in response to your (and other reviewers') feedback. For instance:
> - We moved the previous Figure 1 to the conclusion section (now Figure 2), to avoid the case of reference preceding definition.
> - We revised the connective paragraphs of sections and subsections to align better with the new specification of the three components.
> - We added methodological clarification in the caption and content of Table 2.
> - We expanded Broader Impact Statement to explain the real-world impact that we hope to achieve.
> - We made writing improvements to clarify the logical flow in individual subsections, and included around ten more peer-reviewed references to support our claims.
>
> We recognize that due to time and space constraints, the current changes may not fully address some of your concerns, specifically per-row and per-columns synthesis/critical discussions of works as presented in Table 2. We agree that doing so is highly valuable, however, and have included it in our agenda for future work.

---

### Review · Reviewer_f9At · 2025-11-27

**Summary Of Contributions:**

Summary

This paper organizes different research efforts under the broad theme of 'AI influence'. It proposes AI influence as a research field, distinguishing it from AI safety or AI ethics, and discusses how AI affects human cognitive process of knowledge acquisition, formation of beliefs and judgments. It describes the mechanisms, amplifiers, and consequences of AI influences. It synthesizes a broad, cross-disciplinary literature and offers an organizing framework and a methodological overview (via a large classification table) to catalyze more systematic study and early interventions. The work is a perspective/survey piece aiming to articulate a shared agenda rather than to contribute new empirical or theoretical results.

Strengths
- The three-level framework (mechanisms → amplifiers → consequences) is a useful, high-level organizing lens that helps connect otherwise siloed research communities and phenomena.
- The paper foregrounds feedback-loop dynamics (human-to-AI and AI-to-human) and institutional amplification, which are often treated separately; this cross-pollination is valuable.
- The high-level narrative is clear and accessible, especially for an interdisciplinary audience.
- The methodological survey table gives readers an accessible map of evidence types and where causal or quasi-experimental work exists.
- The sections on mechanisms, amplifiers, and consequences are well-structured and easy to follow at an overview level.

Weaknesses
- The framework remains largely conceptual; there is no formal model or unifying operationalization that renders the “orthogonality” claim testable, nor a set of falsifiable predictions linking mechanisms, amplifiers, and specific consequences.
- The paper does not propose concrete measurement protocols or standardized metrics for “AI influence” beyond citing diverse prior work; a stronger contribution would include an evaluative schema (e.g., metrics, designs, indicators) others can adopt.

**Audience:**

Yes

**Audience Explanation:**

TMLR's ML-focused audience includes researchers interested in AI's societal implications, such as alignment, fairness, and human-AI interaction. The paper's framework could inform work on model biases, recommender systems, or RLHF, aligning with growing discussions on AI's epistemic risks. Even if speculative, it highlights gaps relevant to those studying long-term AI impacts.

**Broader Impact Concerns:**

The agenda is important: as AI becomes integral to information mediation and production, epistemic and value-level impacts merit sustained study. The cross-disciplinary framing is a strength as it addresses an important and timely problem of the epistemic and axiological impacts of AI and offers a clear, accessible framework that can help organize a fragmented literature. On the other hand, the paper's focus on potential harms (e.g., knowledge collapse, epistemic stratification) could potentially influencing policy or public perception negatively.

**Claims And Evidence:**

No

**Claims Explanation:**

Though not an empirical paper, it assembles a broad array of empirical findings across HCI, social science, and ML to support the conceptual framing. However, while the paper cites extensively, many are to unpublished preprints or hypothetical works. Core claims about mechanisms and consequences are often speculative and without robust causal evidence or quantitative modeling. Clearer evidence from established studies (e.g., on biases or offloading) supports some points, but overall, the framework feels more conceptual than convincingly substantiated. Large-scale observational studies provide concrete evidence of AI-mediated shifts in human behavior. Integrating these as case studies would strengthen the empirical grounding and demonstrate how to operationalize “AI influence.”

**Requested Changes:**

- Operationalize “AI influence” with measurable constructs (e.g., belief shifts, framing distributions, concept diversity indices) and propose a core evaluation suite that researchers can adopt across domains. (critical for acceptance)
- Replace or supplement references with more existing, peer-reviewed sources to strengthen evidence. (would strengthen)
- Expand empirical validation, e.g., via case studies or simulations of the framework. (would strengthen)
- Translate the framework into measurement-and-intervention playbook with concrete metrics, study designs, and governance levers. (critical)
- Include a taxonomy of risks with thresholds/leading indicators to enable early warning and practical mitigation by labs and platforms. (critical)
- Provide clearer guidance for ML practitioners (e.g., data workflows that minimize collapse risk; auditing framing and rhetorical density vs. factuality; guardrails for preference-learning feedback loops). (strengthen)

---

> ### Author Response · Authors · 2025-12-12
> **Summary of Changes**
>
> Thank you so much for your feedback!
>
> In our newly revised manuscript, in addition to targeted improvements, we have made the following large structural changes. The brackets preceding each point link it to the changes you have recommended.
> - [1,4] Newly created **Section 2** that specifies the scope of systems we consider, formally explains the absence of AI Influence in the formalisms of these systems, and decomposes the impact of such missing influence into three components, each corresponding to mechanisms, amplifiers, or consequences.
> - [1] Newly created **Figure 1** that illustrates the precise and formal distinction between mechanisms, amplifiers, and consequences, and the reason why they form a complete picture.
> - [1,4] Newly created **Table 1** that compares the formalism of language models training (supervised/reinforced), RecSys training, and the construction of knowledge-based systems. They serve as canonical examples in the scope of systems that we consider.
> - Moved "human-AI feedback loop" from mechanisms to amplifiers. The new formal distinction between the three components resolved the ambiguities and made it clear where it belongs.
>
> We have also made targeted improvements in response to your (and other reviewers') feedback. For instance:
> - [2] We included around ten more peer-reviewed references to support our claims.
> - We revised the connective paragraphs of sections and subsections to align better with the new specification of the three components.
> - We added methodological clarification in the caption and content of Table 2.
> - We expanded Broader Impact Statement to explain the real-world impact that we hope to achieve.
> - We made writing improvements to clarify the logical flow in individual subsections.
>
> We especially liked the changes 5/6 (practical implications) that you requested, and are currently working on them. We noticed that many preprints that we cited have peer-reviewed versions, and are doing batch replacement of those citations, as per your requested change 2. We will follow up upon completion of these tasks; thank you again for the help!

---

### Review · Reviewer_vDA2 · 2025-11-28

**Summary Of Contributions:**

This paper studies how modern AI systems (e.g., LLm), influence human belief formation, truth-seeking ability, and moral development. Its main contribution is a framework that identifies and analyzes the main mechanisms by which AI systems can exert such influence.
First, the authors introduce 3 archetypal mechanisms:
- Direct influence, where AI modifies a user’s internal preferences or mental states
- Information-control influence, where AI systems shape what information is accessible or highlighted for a user
- Behavior-control influence, where AI affects a user’s environment or decision context to steer behavior
Then, the authors also propose a general model of AI influence that unifies these mechanisms and provides criteria for evaluating their potential impact (both beneficial and harmful) on cognition and morality. Lastly, the paper outlines policy considerations and safeguards for mitigating undesirable influence while preserving beneficial uses of AI systems.

Strengths:
- The paper provides a clear conceptual decomposition of influence mechanisms, which helps separate “what AI can do” from “whether AI should do it”
- It offers a structured approach that could guide future theoretical or empirical works
- The unified influence framework may be useful for policymakers and system designers

Weaknesses:
- The work is primarily conceptual and would benefit from empirical validation
- Some mechanisms are described at a high level, which may limit immediate impact on real-world systems
- Not all claims in the paper are supported with relevant citations
- The writing of the paper could be improved

**Additional Comments:**

The “Caveats and Counterarguments” Section (5) raises several worthwhile considerations, but in its current form it reads more like a broad disclaimer than a focused analytical critique. Many of the listed caveats are not integrated into the main argument and tend to dilute rather than clarify the paper’s contributions

**Audience:**

Yes

**Audience Explanation:**

The paper addresses topics, e.g., AI influence, behavioral effects of language models, and mechanisms of persuasive or manipulative impact, that are increasingly relevant to the ML community. Researchers in areas such as alignment, human–AI interaction, safety, ethics, and evaluation would likely find the paper’s findings valuable. The work considers how model outputs shape user beliefs and behavior, which is currently a core concern within the field.

**Broader Impact Concerns:**

The paper includes a Broader Impact Statement, but it is very brief and does not sufficiently engage with the ethical implications of research on AI-driven influence. Since the work touches on sensitive topics, e.g., human behavior, susceptibility to persuasion, and mechanisms of AI influence, it would benefit from a more thorough discussion of potential risks (e.g., misuse for targeted persuasion, societal manipulation, amplification of biases) as well as possible safeguards or mitigations. Expanding this section would strengthen the ethical framing of the paper

**Claims And Evidence:**

No

**Claims Explanation:**

While the paper raises interesting hypotheses about mechanisms by which AI systems may influence human moral and epistemic development, the claims are largely speculative and not grounded in empirical evidence. Several core assertions (e.g., the existence of distinct “influence mechanisms” or the presumed long-term behavioral effects), are presented without experimental validation, user studies, or references to established findings in psychology, cognitive science, or human-computer interaction.
Many sections use anecdotal reasoning or highly abstract arguments instead of rigorous evidence. For example:
- The paper claims that “AI systems consistently shape people’s mental models over time” but provides no longitudinal data, surveys, or controlled experiments to justify this
- The proposed mechanisms are introduced as definitive categories, yet the taxonomy is not derived from data, prior literature, or clearly articulated methodology
- Several normative claims (e.g., about how AI “should” guide moral development) are presented without supporting philosophical argumentation or citations

Because the work lacks empirical results or reproducible studies, the evidence currently provided does not substantiate the paper’s stronger claims. Thus, while the conceptual direction is interesting, the submission does not meet the evidentiary expectations of TMLR

**Requested Changes:**

Critical Changes (must be addressed for acceptance)
- Several foundational terms (e.g., “influence mechanisms,” “truth-seeking,” “moral development,” “autonomy degradation”) are used without rigorous operational definitions. The authors must provide precise, testable definitions and explain how they differ from existing constructs in psychology, media studies, or ethics
- The paper makes several strong causal or quasi-causal claims (e.g., about cognitive manipulation, long-term effects on moral development, or cross-cultural psychological shifts) without empirical backing. The authors must add evidence, empirical studies, or well-established theoretical frameworks that support these claims
- The proposed taxonomy appears high-level but lacks a clear methodology for how it was constructed. Please provide a systematic process (e.g., grounded theory, literature synthesis workflow, coding schema, expert validation) so readers can evaluate its scientific validity
- Some concepts are used inconsistently across the paper (e.g., “manipulation,” “nudging,” “influence,” “guidance”). Ensure terms are consistent and contradictions removed, otherwise the framework becomes ambiguous
- Improve the writing clarity in several sections: Some paragraphs are difficult to follow due to grammar issues, vague wording, or overly abstract phrasing. The paper needs a comprehensive language and structure edit to ensure clarity and readability
- The paper currently emphasizes risks, but actionable mitigation strategies (auditing, transparency, agent behavior constraints, etc.) are not explored in sufficient depth. Adding these would greatly improve the practical relevance

---

> ### Author Response · Authors · 2025-12-12
> **Summary of Changes**
>
> Thank you so much for the informative feedback!
>
> In our newly revised manuscript, in addition to targeted improvements, we have made the following large structural changes. The brackets at the beginning map them to the critical changes you have recommended.
> - [1,3] Newly created **Section 2** that specifies the scope of systems we consider, formally explains the absence of AI Influence in the formalisms of all these systems, and decomposes the impact of such missing influence into three components, each corresponding to mechanisms, amplifiers, or consequences.
> - [1,3,4] Newly created **Figure 1** that illustrates the precise and formal distinction between mechanisms, amplifiers, and consequences, and the reason why they form a complete picture.
> - [3,5] Newly created **Table 1** that compares the formalism of language models training (supervised/reinforced), RecSys training, and the construction of knowledge-based systems. They serve as canonical examples in the scope of systems that we consider.
> - [3] Moved "human-AI feedback loop" from mechanisms to amplifiers. The new formal distinction between the three components resolved the ambiguities and made it clear where it belongs.
>
> We have also made targeted improvements in response to your (and other reviewers') feedback. For instance:
> - [5] We revised the connective paragraphs of sections and subsections to align better with the new specification of the three components.
> - [2,4] We have added methodological and taxonomy clarifications in the content and caption of Table 2, and added references to works in the table for empirical support of claims.
> - We expanded Broader Impact Statement to explain the real-world impact that we hope to achieve.
> - [5] We made writing improvements to clarify the logical flow in individual subsections.
>
> We recognize that some of your concerns are not fully addressed by the changes outlined above, and require additional structural/systematic change. We are working on such changes at the moment, and will follow up upon completion. Thank you again for the help!

---

### Review · Reviewer_X7UB · 2025-11-29

**Summary Of Contributions:**

The authors define a new concept, AI Influence, and try to explain and clarify it from three aspects: Mechanisms, Amplifiers, and Consequences.

**Strengths**:

1. The authors introduce several new concepts and review relevant literature.

2. As the authors have pointed out, within the scope of "AI influence" as defined by them, there is indeed a lack of papers that summarize some related phenomena.

**Weaknesses**:

1. The authors attempt to define the new concept of "AI influence", but do not provide a clear discussion of the related terminology. No papers have used this term (even if with a different meaning)? Which have used similar ones? What are the distinctions?

2. The title is too broad relative to the content. The authors seem to equate LLMs with AI sometimes, but LLMs are just one type of AI and do not represent all of AI models.

3. The classifications in the article lack logic, such as the selection of subcategories under "Mechanisms", "Amplifiers", and "Consequences".

4. The authors' selection of literature is unbalanced. For instance, some categories in Table 1 list many items, while others have very few or are even left blank. In addition, there is a lack of references to earlier literature.

**Audience:**

Yes

**Audience Explanation:**

Ethical issues and social impacts need to be considered in the development of AI. This article could inspire further thought among researchers.

**Broader Impact Concerns:**

Given the topic of the article, the Broader Impact Statement, which is currently only one sentence, could be expanded.

**Claims And Evidence:**

No

**Claims Explanation:**

1. Many of the claims made by the authors are neither precise nor clear. For example, the article lacks supporting data, and some arguments are fragmented.

2. The motivation for the classification shown in Figure 1 is unclear, and it feels like a patchwork. For instance, the first item "Digital Reliance" listed by the authors in "Mechanisms" could also be classified in "Amplifiers" or "Consequences". Why choose "Feedback Loop from Dual Influence" and "Goodhart's Law and Value Capture"?

3. Table 2 is also not well organized. For example, the authors treat RCTs and causal inference as two distinct methods. Why is "Qualitative Research" listed but "Quantitative Research" not considered?

4. Some statements are exaggerated or lack references, for example,

> We summarize all the methods that are used so far to study AI influence in Table 1.

> Human objectives are often operationalized into quantifiable metrics

**Requested Changes:**

1. The authors need to narrow the scope of the vision in the title or include more necessary references.

2. The authors need to review the survey articles related to "AI influence" (not necessarily using this term) and clarify how these articles analyze "AI influence" (i.e., from which perspectives).

3. The classifications in Figure 1 and Table 1 need to clearly outline the motivation and rationale. Appropriate adjustments, such as splitting, merging, or renaming, should be made.

4. More figures and tables are needed to provide clarification. It is unreasonable to rely on just one figure and one table for such a broad topic.

---

> ### Author Response · Authors · 2025-12-12
> **Summary of Changes**
>
> Thank you so much for the helpful feedback! In our newly revised manuscript, in addition to targeted improvements, we have made the following large structural changes:
> 1. Newly created **Section 2** that specifies the scope of systems we consider, formally explains the absence of AI Influence in the formalisms of all these systems, and decomposes the impact of such missing influence into three components, each corresponding to mechanisms, amplifiers, or consequences.
> 2. Newly created **Figure 1** that illustrates the precise and formal distinction between mechanisms, amplifiers, and consequences, and the reason why they form a complete picture.
> 3. Newly created **Table 1** that compares the formalism of language models training (supervised/reinforced), RecSys training, and the construction of knowledge-based systems. They serve as canonical examples in the scope of systems that we consider.
> 4. Moved "human-AI feedback loop" from mechanisms to amplifiers. The new formal distinction between the three components resolved the ambiguities and made it clear where it belongs.
>
> We have also made targeted improvements in response to your (and other reviewers') feedback. For instance:
> - We revised the connective paragraphs of sections and subsections to align better with the new specification of the three components.
> - We renamed "causal inference" as a methodology to "observational causal inference", to clarify that we only include non-interventional studies based on observational data. We also added methodological clarification in the table caption.
> - We expanded Broader Impact Statement to explain the real-world impact that we hope to achieve.
> - We made writing improvements to clarify the logical flow in individual subsections.
>
> Some remaining changes you recommended require more time to make. We are working on them at the moment, and will follow up upon completion. Thank you again for the help!

---

### Decision · Action_Editor_bzmM · 2026-01-01

**Recommendation:** Reject

**Additional Comments:**

The recommendation is based on the reviewers' comments, the action editor's evaluation, and the authors’ response.

This submission should not be accepted in its current form due to several fundamental issues, as pointed out by the reviewers. All reviewers recommend rejecting the current version. The main weaknesses summarized from reviewers' comments are listed below.

Reviewer EsgC
- Section 2 on AI Influence opens with "intelligent systems are designed to interact with humans in ways that alter human cognition. [...] Current examples of such systems include:" and then describes LLMs as interactive tools. I do not understand how that "alters human cognition". The other two examples (recommenders and decision support system) are somehow clearer but still "altering human cognition" seems rather inprecise.
- I had criticized some cited research as wrongly framed, such as Wahn et al. (2023) in section Cognitive offloading, but nothing has changed in this section, unfortunately.
- Now comparing the new version with the older version carefully, most of the text actually remained the same, such as the socio-economic effect, knowledge collapse, epistemic stratification, and so on, for which the reviewers requested much more evidence and clarity of arguments and being less speculative.

"In this new section, we have also aimed to unambiguously lay out what our "position" is."

- Section 2 now tries to formalize AI influence (somehow through some sort of contitioned probability in Figure 1 (??) but the unambiguous position of this paper still remains unclear to me. Why should we bother about coining a new umbrella term for "AI influence" in the first place? I'm having a hard time to get the main gist of the paper, which is somehow between a survey, speculation, and (partly) formalization.

 Reviewer X7UB
- Although certain improvements have been made, the fundamental issues have not been addressed. The current analysis and the paper still fail to support the title and conclusions, and more work is needed on the literature review.

 Reviewer vDA2
- The revised version does not adequately address the issues raised by the reviewers. Moreover, the newly added formulation does not meaningfully contribute to the paper or substantiate any of its claims; instead, it appears somewhat forced and artificial.

Reviewer f9At
- The claims are largely speculative and without formal operationalization or falsifiable predictions.

**Audience:**

Yes

**Audience Explanation:**

The topic is of general interest to TMLR readers

**Claims And Evidence:**

No

**Claims Explanation:**

Multiple reviewers noted that many key claims lack sufficient evidence to support them.

**Resubmission Of Major Revision:**

The authors may consider submitting a major revision at a later time.